# Improving HIV stigma in the marginalized population in Khorramabad, Iran: A single-blinded randomized, controlled educational trial using role-playing and lecturing

Mina Jomezadeh[1], Fereshteh Zamani-Alavijeh[2], Forugh Aleebrahim[1], Maryam Nasirian[3]*

1 Student Research Committee, School of Health, Isfahan University of Medical Sciences, Isfahan, Iran,
2 Department of Health Education and Promotion, School of Health, Isfahan University of Medical Sciences, Isfahan, Iran, 3 Epidemiology and Biostatistics Department, Health School; and Infectious Diseases and Tropical Medicine Research Center, Isfahan University of Medical Sciences, Isfahan, Iran

* maryamnasirian17@gmail.com

**Data Availability Statement:** In order to conduct our study entitled "Improving the HIV-stigma in the Marginalized population in Khorramabad, Iran: a

## Abstract

This study aimed to investigate the effect of role-playing and lecture on improving the attitudes of the Khorramabad suburban population toward the human immunodeficiency virus (HIV). In 2019, 270 people aged 18 and above participated in a randomized controlled trial in Khorramabad, Iran. Individuals were randomly assigned to one of three groups: role-playing, lecture, or control. To collect data before and after the intervention, we used a standard HIV knowledge and attitude questionnaire. Before the educational intervention, three marginalized groups' attitudes toward HIV were stigmatized. After the intervention, the attitudes of both the role-playing and lecture groups improved significantly relative to the control group (P<0.0001); there was no significant difference between the two methods (P>0.05). The correlation between attitude and knowledge scores was positive (P<0.0001). This study demonstrates that education is a fundamental pillar of improving attitudes toward people living with HIV (PLHIV) and can reduce stigma against them, thereby increasing their desire to disclose their condition and seek medical care.

**Trial registration:** The trial registration code is IRCTID: IRCT20190807044467N1 (https://en.irct.ir/trial/41464).

## Introduction

The World Health Organization (WHO) estimated that by 2019, 38 million people would be infected with human immunodeficiency virus (HIV), including 1.7 million new cases and 690,000 deaths due to acquired immunodeficiency syndrome (AIDS). According to a report published by the Iranian Ministry of Health, there were 41494 people infected with HIV and 19164 deaths as of the end of 2019 [1]. In addition, approximately 1293 cases were reported in the Iranian province of Lorestan. According to a 2017 study conducted in this province's

single-blinded randomized, controlled educational trial using role-playing and lecturing", in addition to receiving the Ethical code (IR.MUI.RESEARCH. REC.1398.482) and the RCT code (IRCT20190807044467N1), we had to obtain an executive permission from the Centers for Disease Control (CDC), the Ministry of Health (MOH). In the received permission, it is officially stated that we are not allowed to publish individual data. It is also mentioned that if we want to publish information, we should coordinate with CDC. It should be noted that the data set is now kept confidential by the project manager and the corresponding author of the article, Dr. Maryam Nasirian. But to receive data, permission must be obtained from the Dr. Hengameh Namdari (namdarih@gmail.com), the manager of Department of AIDS and Sexually Transmitted Infections, CDC, MOH. We confirm that any researcher can obtain the data set in the same way that we have obtained it.

**Funding:** The authors received no specific funding for this work.

**Competing interests:** The authors have declared that no competing interests exist.

capital, Khorramabad, the number of HIV-positive individuals was estimated to be 2,456 [2], and some of these individuals were observed to reside in three marginalized regions surrounding Khorramabad; however, no accurate data regarding the number of patients in these areas exist.

HIV infection is a threat to marginalized populations. High population density, lack of adaptability and coexistence between different cultures, increasing unemployment and low income, lack of adequate welfare and educational facilities, relatively low level of public health, insufficient literacy, and lack of knowledge about HIV, as well as negative attitudes toward patients with HIV, provide a foundation for the spread of high-risk behaviors and the prevalence of HIV in these populations [3–5]. Consequently, planning for HIV prevention and control in this vulnerable population appears necessary.

HIV/AIDS remains a serious problem in the global health system due to its diverse modes of transmission, high mortality rate, high cost of treatment, and social issues such as stigma and discrimination. To this end, HIV prevention methods remain one of the WHO's primary concerns. However, it is essential to consider that HIV infection affects patients' mental and social health and physical health, owing primarily to society's negative attitude toward the disease. Negative attitudes lead to the stigmatization of patients and, as a result, their discrimination and social exclusion; consequently, many patients attempt to conceal their illness, even from their partners or medical professionals [6].

According to statistics in Iran, approximately 15.4% of people who living with HIV (PLHIV) fail to visit service centers to receive care. In the Lorestan province, approximately 71% of the total number of registered cases are active, and about 80% of active cases receive treatment [7]. The unwillingness of patients to disclose their disease, on the one hand, results in inadequate health care services and, on the other, the spread of infection among their acquaintances and other members of society. Due to a lack of knowledge about HIV, stigmatizing attitudes exist [8]. To this end, reducing the stigmatizing attitude towards HIV patients, particularly vulnerable and high-risk populations, is an important method for preventing the spread of HIV, where only possible through HIV education and awareness [9].

There are several HIV education strategies, but the two most prevalent and influential are lecture and role-playing, each of which has advantages and disadvantages [10]. The advantages of the lecture method are a large number of students relative to the teacher, its low cost, the strengthening of social relationships, and increasing the self-confidence and interest of the students [11]. The superior characteristics of the role-playing method are that the audience develops an emotional connection with the performance and the role-players and watches the play's development with excitement and a sense of being onstage. As a result of the method's emphasis on the senses and emotional communication, learning is enhanced and made more efficient [12].

Due to the high prevalence of HIV infection in Khorramabad, particularly in the marginalized regions [13], and the lack of adequate education for people in these regions, the present study aimed to examine the effects of two educational methods, lecture, and role-playing, on reducing stigmatizing attitudes towards patients as a key factor in controlling and preventing the spread of HIV among the vulnerable population.

## Material and methods

### Study design, participation, and sampling

The current study was a single-blinded, parallel, randomized, controlled educational trial study comprising three education groups: 1) role-playing, 2) lecture, and 3) a control group (Fig 1).

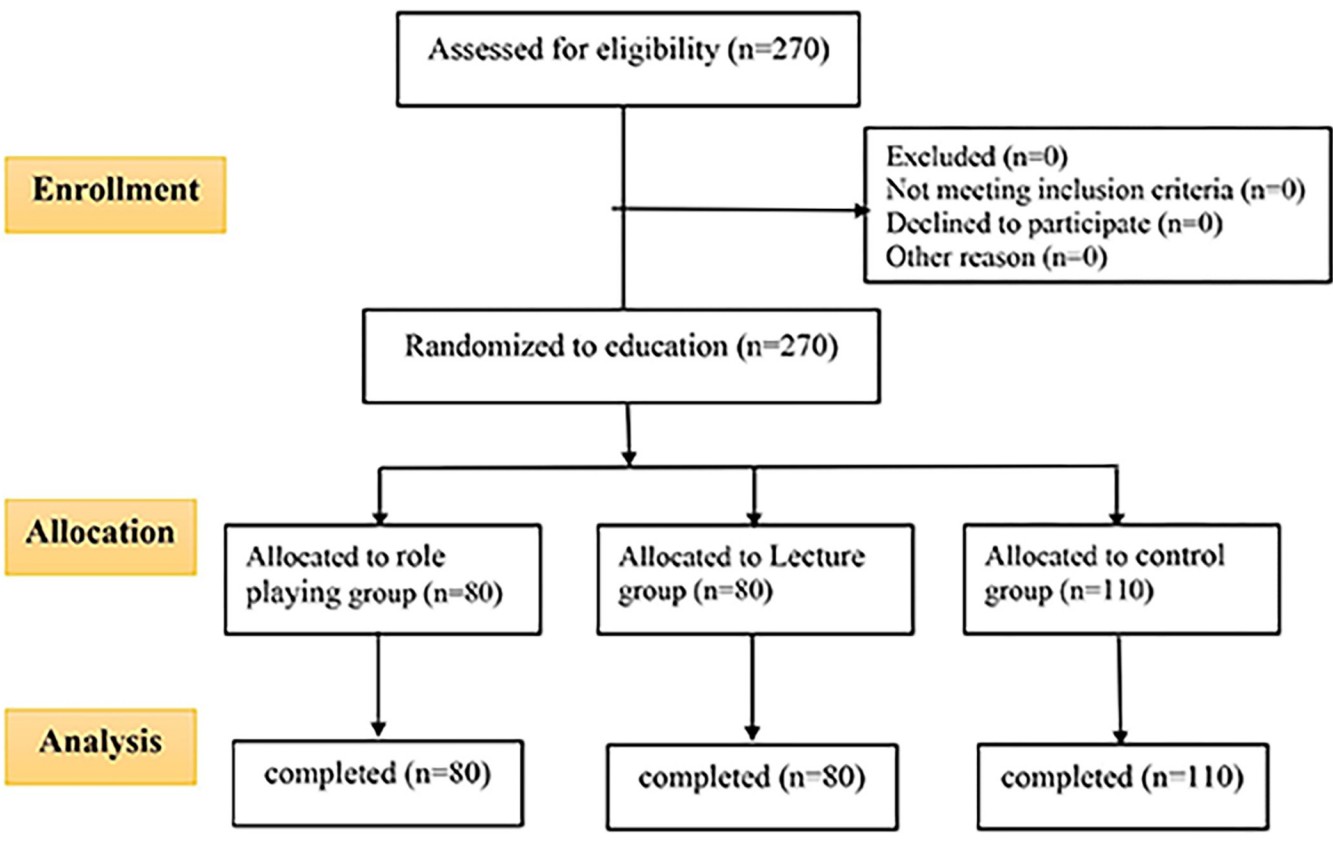

**Fig 1. CONSORT flow diagram of participant attitudes.**

This study aimed to examine the effects of two educational methods, lecture and role-playing, on reducing stigmatizing attitudes toward patients, a key factor in controlling and preventing the spread of HIV in the vulnerable population.

The present study was conducted as a three-armed randomized controlled educational trial from May to July 2019 at Isfahan University of Medical Sciences on a marginalized population aged 18–49 living in three regions (a total population of 38731); Falak od Din (N = 11313), Masur (N = 12475), and Poshteh (N = 14943) villages in Khorramabad city. The sample size was calculated using the formula for multi-arm clinical trial studies and considering the differences in mean attitude scores between the role-playing group (95.4) and the lecture group (98.3) based on a method described in Abedian et al. [10], a confidence level of 95%, 80% test power, and a 10% probability of loss during the study, equal to 110 in the control group and 80 per intervention group (lecture and role-playing).

Each region had three health centers, which were considered classes, and sampling was performed based on the population covered by each center, the share of age and gender groups, and the existing list of individuals on the Sib website using simple random sampling. After matching against the inclusion and exclusion criteria, the individuals were contacted by phone and invited to visit the relevant centers to participate in the study. If an individual declined to participate or were ineligible, another person of the same age and gender would be randomly selected and contacted by phone. The procedure was repeated until samples were collected from all age and gender groups.

Inclusion criteria for this study included: age between 18 and 49, Iranian citizenship, residence in Falak od Din, Masur, and Poshteh, elementary school education or higher, mental capacity to comprehend education and produce a response, absence of known psychiatric disorders or severe visual and hearing impairments, and written informed consent to participate in the study. They were also excluded from the study if they refused to participate or died during the research.

To meet the requirement of blinding the participants, they were randomly assigned by region because it was impossible to blind individuals from the same region, and despite the intention, they were aware of which group they belonged to after speaking with other participants. Therefore, we randomized the population of Falak od Din to participate in the role-playing group, the population of Poshteh to receive lectures, and the population of Masur to serve as a control.

## Research instruments

We collected data using a standard questionnaire. In a study conducted by Tavakoli, the content and face validity of the questionnaire were evaluated using a qualitative method, and Cronbach's alpha for internal validity was 0.79 for the attitude section and 0.77 for the knowledge section [14]. The main section of the questionnaire consisted of 18 questions (4 items) about the attitude assessment. The remaining two sections contained 24 knowledge-based questions and five questions about demographic data.

The stigmatizing attitude was measured on a 5-point Likert scale (strongly disagree, disagree, neutral, agree, strongly agree), with 14 negative questions scored from 1 (most positive attitude) to 5 (most negative attitude) and four positive questions scored in reverse order. Therefore, the range of scores obtained from all attitude questions was between 18 (positive attitude) and 90 (stigmatizing attitude), with the highest score indicating a more stigmatizing attitude. Consequently, participants who scored above the mean (54 points) were categorized as having stigmatizing attitudes toward patients.

Yes and no answers were required for the knowledge section, with a score of 1 for each correct response and a score of 0 for each incorrect response. The scores obtained from the total number of attitude questions ranged from 0 to 24, and the lowest score indicated a lack of knowledge.

## Data collection

The educational material was extracted from the booklet titled "a new approach to HIV education," compiled by the Department of AIDS and Sexually Transmitted Diseases within the Ministry of Health and Medical Education [15]. In both educational approaches, HIV biology, epidemic conditions in the province and Iran, transmission methods, high-risk behaviors, self-preservation skills, and negative attitudes and misconceptions about HIV patients were covered. The educational intervention was conducted in two groups, "lecture" and "role-playing," and a pamphlet was designed and distributed to both groups. The control group received no intervention.

The participants provided written informed consent and completed the self-administered knowledge and attitude questionnaire after explaining the purpose of the research and assuring them of the confidentiality of their data. Afterward, the interventions, which included teaching via lecture and role-playing, were performed according to the grouping, and the educational schedule was conducted for one hour per week for one month in the marginalized regions' health community centers and mosques.

Due to the extremely high possibility of displacement and disappearance of individuals in marginal populations, the interval between the conclusion of educational interventions and

the reevaluation of individuals was determined to be two weeks, and the questionnaire was administered to all individuals in the control group and both intervention groups again.

## Data analysis

We analyzed the data using SPSS V.23 and characterized them using the mean, standard deviation, frequency, and percentage. In addition, we analyzed the data using the independent two-sample t-test, analysis of variance (ANOVA), and Chi-square test at a significance level of 5%. Multiple analyses were used to examine the effects of group differences before the intervention, confounding variables, and the correlation between knowledge and attitude scores. The significance level in all analyses was 5%.

## Ethics approval

The Iranian Registry of Clinical Trials approved all experimental protocols with the code IRCT20190807044467N1. In addition, Iran National Committee for Ethics in Biomedical Research approved this study with an ethical code (IR.MUI.RESEARCH.REC.1398.482), and Isfahan university of medical science approved the proposal with code 398553.

## Trial protocol

The trial protocol is available at https://en.irct.ir/trial/41464.

## Results

We included a total of 270 individuals (51% females) between the ages of 18 and 49 from marginalized regions, Falak od Din (role-playing: 80), Masur (control: 110), and Poshteh (lecture: 80). There was a statistically significant difference (p = 0.0190) in the mean age of the participants across the three groups, but there were no differences (P>0.05) in any of the other demographic variables. The lecture group had a significantly longer history of receiving HIV education than the other two groups (p<0.0001) (Table 1).

The overall results indicated that before the intervention, the average knowledge score in people living in Khorramabad's suburban areas was around 17.4 (4.4). Furthermore, before the intervention, the control group's mean knowledge score was significantly lower than that of both the lecture and role-playing groups (p = 0.001), while there was no significant difference between the lecture and role-playing groups (p = 0.972). After controlling for the effects of the age difference and history of HIV education in the groups before the intervention, the mean post-intervention knowledge scores were significantly different between groups, with the control group scoring significantly lower than the intervention groups (p<0.0001). There was, however, no statistically significant difference between the lecture and role-playing groups (p = 0.650). Despite a significant increase in knowledge scores in all three groups due to the intervention, the increase in the control group was significantly different from the increases in the intervention groups (p<0.0001). Before the intervention, there was a significant correlation between attitude and knowledge scores (r = 0.425, p 0.0001), and after the intervention, the correlation remained significant (r = 0.484, p<0.0001) (Table 2).

Overall, the results showed that the average score of individuals' attitudes in marginalized areas of Khorramabad was approximately 44.4. This value was near the median (54), indicating that their attitudes toward HIV were stigmatized. Furthermore, at the start of the study, approximately 15.6% of all participants had stigmatizing attitudes (a score over 45). The stigmatizing attitude was significantly lower in the lecture group than in the control group (p = 0.012), but there was no significant difference in the role-playing group (p = 0.262). After

**Table 1. Demographic characteristics of the participants.**

| Variable | Study Groups | | | P-Value |
|---|---|---|---|---|
| | Lecture n = 80 | Role playing n = 80 | Control n = 110 | |
| **Mean-age (Sd)** | 32.87(8.41) | 33.36(7.24) | 32.42(9.75) | 0.019* |
| **Gender-n(%)** | | | | |
| Female | 41.(51.25) | 42(52.50) | 50(55) | 0.943^ |
| Male | 39(48.75) | 38(47.50) | 50(55) | |
| **Marital Status** | | | | |
| Single | 22(27.50) | 16(20.00) | 28(25.45) | 0.590^ |
| Married | 58(72.50) | 64(80.00) | 82(74.55) | |
| **Educational Status** | | | | |
| Elementary | 3(3.75) | 4(5.00) | 7(6.36) | 0.821^ |
| Under diploma/Diploma | 49(61.25) | 55(68.75) | 68(61.82) | |
| Educated | 28(35.00) | 21(26.25) | 35(31.82) | |
| **History of receiving HIV education** | 43(53.75) | 20(25.00) | 40(36.36) | 0.0001^ |

* One-way Anova

^ Chi-square

p-value is significant at 0.05

controlling for age differences, history of HIV education, and attitude score before the intervention, the mean scores of stigmatizing attitudes in the lecture and role group groups were significantly lower after the intervention than in the control group ($p<0.0001$); however, no meaningful difference was observed between the two groups (p = 0.623). After controlling for differences in age, history of HIV education, and attitude score, the mean scores of negative attitudes toward all components were significantly lower in the lecture and role-playing groups than in the control group ($p<0.0001$). Furthermore, in the lecture and role-playing groups, there was a substantial reduction in negative attitude scores and all of its components after the intervention compared to before the intervention, whereas the changes were not meaningful in the control group (Table 3).

## Discussion

The present study, which examined the marginalized population between the ages of 18 and 49 in Khorramabad, revealed that educational interventions could positively affect the marginalized population's attitude toward PLHIV. Despite the lack of research on the effect of

**Table 2. Comparison of the knowledge between the groups before and after the intervention.**

| Knowledge score | Mean scores (Sd) | | | ANOVA/ MANOVA* | |
|---|---|---|---|---|---|
| | Lecture n = 80 | Role playing n = 80 | Control n = 110 | F | P-value |
| **Before the intervention** | 18.3(3.8) | 18.1(4.2) | 15.9(6.7) | 9.12 | <0.0001 |
| **After the intervention** | 24.9(2.3) | 23.4(2.0) | 18.1(4.5) | 10.56 | <0.0001 |
| **Paired t test value** | -13.61 | -14.80 | -5.93 | | |
| **P-value** | <0.0001 | <0.0001 | <0.0001 | | |

* MANOVA was used for group comparisons after intervention while adjusted for the differences between the groups before the intervention in terms of history of receiving HIV training and age

**Table 3. Comparison of the attitude and its components between the groups before and after the intervention.**

| Component | Mean Score (Sd) | | | ANOVA/ MANOVA* | |
|---|---|---|---|---|---|
| | Lecture n = 80 | Role playing n = 80 | Control n = 110 | F | P-value |
| **Total Score of Attitude** | | | | | |
| Before the intervention | 42.2(8.1) | 44.5(8.4) | 45.9(9.8) | 4.25 | 0.0153 |
| After the intervention | 30.6(7.8) | 32.1(6.7) | 46.7(9.5) | 4.00 | <0.0001 |
| Paired t test value | 9.24 | 12.5 | -0.95 | | |
| P-value | <0.0001 | <0.0001 | 0.367 | | |
| **Patients' social status Score** | | | | | |
| Before the intervention | 14.9(4.1) | 15.7(3.7) | 14.44(5.1) | 1.92 | 0.148 |
| After the intervention | 10.1(3.4) | 10.2(2.9) | 14.94(4.8) | 2.74 | <0.0001 |
| Paired t test value | 10.08 | 11.98 | -1.31 | | |
| P-value | <0.0001 | <0.0001 | 0.189 | | |
| **Patients' social support score** | | | | | |
| Before the intervention | 8.8(2.8) | 9.3(3.4) | 9.9(3.8) | 2.61 | 0.084 |
| After the intervention | 6.5(2.3) | 7.1(2.2) | 9.8(3.7) | 2.91 | <0.0001 |
| Paired t test value | 5.81 | 5.80 | 0.08 | | |
| P-value | <0.0001 | <0.0001 | 0.936 | | |
| **Social perception of disease score** | | | | | |
| Before the intervention | 14.0(3.9) | 14.7(4.32) | 16.3(5.2) | 6.46 | 0.002 |
| After the intervention | 11.1(3.6) | 11.5(3.1) | 16.4(4.7) | 3.53 | <0.0001 |
| Paired t test value | 5.31 | 6.23 | -0.08 | | |
| P-value | <0.0001 | <0.0001 | 0.935 | | |
| **Patients' social harassment score** | | | | | |
| Before the intervention | 4.4(1.9) | 4.9(1.8) | 5.3(4.9) | 5.27 | 0.006 |
| After the intervention | 3.1(1.2) | 3.4(1.3) | 5.5(5.1) | 4.41 | <0.0001 |
| Paired t test value | 5.32 | 8.71 | -1.07 | | |
| P-value | <0.0001 | <0.0001 | 0.284 | | |

* MANOVA was used for group comparisons after the intervention while adjusted for the differences between the groups before the intervention in terms of history of receiving HIV training and age

education on improving social attitudes toward PLHIV, numerous studies [16, 8] have emphasized the importance of education in reducing HIV-related stigma. The findings of the CHAMP study on racial and ethnic diversity in the United States highlighted the importance of education in reducing stigmatized attitudes toward patients [17].

An intriguing aspect of the current study was that a significant proportion of the participants did not receive AIDS training until the intervention. Assuming that the marginalized population of Khorramabad was representative of all marginalized populations in Iran and considering the populations' susceptibility to high-risk behaviors and, consequently, HIV infection [3–5], it would appear that Iran must plan for the education of these marginalized populations.

According to the results, there was a positive correlation between HIV knowledge and attitude, so a higher level of HIV knowledge improved attitudes toward the disease and patients. Therefore, it appears necessary to increase HIV knowledge as a prerequisite for changing societal attitudes [18]. Notably, a high level of HIV knowledge alone will not change societal attitudes; therefore, it is necessary to design specific educational programs to improve attitudes.

Despite the lack of difference between the effectiveness of role-playing and lecture in improving attitude in the present study, studies [20, 19] observed that role-playing has a greater impact on improving knowledge and attitude. Therefore, we recommend conducting additional research on the use of appropriate educational methods for the emotional domain of learning and shifting attitudes toward HIV, such as multimedia methods, scenarios, role-playing, plays, and group discussions [21,22].

Not every educational method can be used to improve attitudes in marginalized populations, as their cultural, economic, and social circumstances and lifestyles differ from those of the general population. For instance, studies have revealed that the general and health literacy levels of marginalized populations are lower than the national average [3–5]; thus, additional research is required to determine the most effective educational approach for these vulnerable populations.

This study's results revealed that the three groups' attitude scores significantly differed before any educational intervention. According to the studies, the disparity was attributable to the different histories of HIV education between the groups. The group that had received more HIV education in the past had a more positive attitude. Given that the groups did not differ in terms of characteristics such as education levels, this result at the onset of the study indicated the effect of education on improving the attitudes of individuals. However, negative attitudes decreased in both the lecture and role-playing groups following the educational intervention, and the effects of differences in the history of HIV training were controlled, demonstrating the effectiveness of education in improving attitudes in the present study.

The results also indicated that the disparity in the overall attitude scores of the three groups at the outset of the study resulted from the difference in the scores of social harassment and social support. Multiple studies have found that HIV-positive individuals do not receive sufficient social support [23,24]. Therefore, we recommend planning to improve the social support for people with HIV from their families and society and educate individuals to improve their attitudes toward HIV patients.

Notably, at the conclusion of the study, the negative attitude increased in the control group, who did not receive any training, most likely due to a coincidence where news of the sudden discovery of a large number of people living with HIV for unknown reasons prompted people to fear and distrust the health care system. The wave of concern spread throughout Iran, particularly in rural regions and marginalized populations, for several months [25], and the news may have contributed to an increase in stigmatizing attitudes towards HIV in the control group. Negative attitudes decreased despite such conditions in the lecture and role-playing groups, indicating that education positively improves attitudes regardless of the circumstances.

One of the study's limitations was the possibility of different attitudes between those willing and unwilling to participate. To reduce contamination bias in education, we assigned three regions to intervention and control groups instead of participants. Nonetheless, the residents of these regions may interact. Due to the cultural taboo surrounding HIV in Iran, participants may not have answered the questions correctly, and social desirability bias existed. The interval between the conclusion of educational interventions and the reevaluation of individuals was two weeks, but a longer interval would have resulted in a more accurate assessment due to the high likelihood of population displacement among marginalized groups.

## Conclusion

Education must be a pillar of efforts to reduce stigmatizing attitudes toward HIV and its patients. Improving societal attitudes improves social support for HIV patients who are

eventually exposed to the disease and receive medical care, an effective step in preventing and controlling the disease. Given the susceptibility of marginalized populations to HIV and their unique living conditions, additional research should be conducted to identify effective and appropriate educational strategies for improving attitudes in these populations.

## Supporting information

**S1 File. Trial protocol.**
(PDF)

## Acknowledgments

We would like to thank Miss Setayesh Sindarreh and Mr. Hamid Mokhayri for their assistance with this study. In addition, the authors thank the participants for their time and participation in this research. Moreover, we appreciate the approval of our proposal by the Isfahan University of Medical Sciences and Health Services Ethics Committee.

## Author Contributions

**Data curation:** Mina Jomezadeh.

**Investigation:** Mina Jomezadeh, Fereshteh Zamani-Alavijeh, Forugh Aleebrahim, Maryam Nasirian.

**Methodology:** Mina Jomezadeh, Fereshteh Zamani-Alavijeh, Forugh Aleebrahim, Maryam Nasirian.

**Project administration:** Maryam Nasirian.

**Resources:** Mina Jomezadeh.

**Software:** Forugh Aleebrahim, Maryam Nasirian.

**Supervision:** Maryam Nasirian.

**Validation:** Fereshteh Zamani-Alavijeh, Maryam Nasirian.

**Visualization:** Mina Jomezadeh, Forugh Aleebrahim.

**Writing – original draft:** Mina Jomezadeh, Forugh Aleebrahim.

**Writing – review & editing:** Mina Jomezadeh, Fereshteh Zamani-Alavijeh, Maryam Nasirian.

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
