## [Decision Letter · Decision Letter 0]

21 Jul 2022

PGPH-D-21-00852

Improving the HIV-stigma in the Marginalized population in Khorramabad, Iran: a single-blinded randomized, controlled educational trial using role-playing and lecturing

Dear Dr. nasirian,

Thank you for submitting your manuscript to PLOS Global Public Health. After careful consideration, we feel that it has merit but does not fully meet PLOS Global Public Health’s publication criteria as it currently stands. Therefore, we invite you to submit a revised version of the manuscript that addresses the points raised during the review process.

Please note that we have only been able to secure a single reviewer to assess your manuscript. We are issuing a decision on your manuscript at this point to prevent further delays in the evaluation of your manuscript. Please be aware that the editor who handles your revised manuscript might find it necessary to invite additional reviewers to assess this work once the revised manuscript is submitted. However, we will aim to proceed on the basis of this single review if possible. 

We look forward to receiving your revised manuscript.

Kind regards,

Julia Robinson

Executive Editor

Journal Requirements:

1. Please provide a detailed online Financial Disclosure statement. This is published with the article. It must therefore be completed in full sentences and contain the exact wording you wish to be published.

a. Please clarify all sources of funding (financial or material support) for your study. List the grants (with grant number) or organizations (with url) that supported your study, including funding received from your institution. 

b. State the initials, alongside each funding source, of each author to receive each grant.

c. State what role the funders took in the study. If the funders had no role in your study, please state: “The funders had no role in study design, data collection and analysis, decision to publish, or preparation of the manuscript.”

d. If any authors received a salary from any of your funders, please state which authors and which funders.

2. Please update your online Competing Interests statement. If you have no competing interests to declare, please state: “The authors have declared that no competing interests exist.”

3. Please provide separate figure file in .tif or .eps format and ensure that all files are under our size limit of 10MB.

4. We have noticed that you have uploaded Supporting Information files, but you have not included a list of legends. Please add a full list of legends for your Supporting Information files after the references list.

Additional Editor Comments (if provided):

Reviewer #1: Improving the HIV-stigma in the Marginalized population in Khorramabad, Iran: a single-blinded randomized, controlled educational trial using role-playing and lecturing

The authors aimed to illustrate the critical role that education on reducing HIV stigmatising behaviours and support social support for people living with HIV. The study was well designed and the finings confirm that education remains a valuable tool to improve the attitude of marginalised population towards people living with HIV. There are few issues that the authors should consider addressing to improve the quality of their submission. I have outlined some of these in detail below but generally, the authors characteristically write long sentences, which make reading and following them a bit challenging.

Introduction:

The authors wrote: "According to the report of the Ministry of Health, there were 41494 people infected with HIV by the end of 2019, and deaths of 19164 people were recorded [1]." It is unclear what Ministry of Health the authors are referring to. Following this, I suggest that the authors should make mention of what country their study is based on because the next sentence goes to mention a province. Not all the readers would be privy with such information and so the authors should consider providing it.

Furthermore, there were about 1293 cases I n Lorestan province, but based on a study by Poorolajal et al. (2017), the number of people living with HIV in the capital of this province, Khorramabad, is probably much higher and is estimated at 2456 ]2[, and it seems that some of them are residents of three marginalized regions around Khorramabad, but unfortunately there is no accurate information about the number of patients in these regions.----This is a very long sentence, the authors should consider breaking this to two or three sentences.

Residents of marginalized regions are vulnerable populations to HIV ---- It is unclear what the authors mean by marginalised regions in this context.

The authors wrote: HIV/AIDS has become an acute problem in the health system worldwide ---- I am not sure if this is accurate. Is HIV an "acute problem". I thought it is more than endemic? The authors should consider reviewing the entire sentence for accuracy.

The authors wrote: "The World Health Organization's main activities seem to focus on treating the patients and preventing infection by controlling the ways of its transmission…" ---- This is not completely accurate; it is only recently with findings showing that undetectable viral load tremendously reduces the chances of HIV transmission. But before this and even up to now there are still predominant HIV prevention methods including most recently the use of PrEP.

According to statistics in Iran, about 15.4% of HIV-infected people do not go to service centers to receive services ---- The authors should consider citing this source.

Sampling

It is unclear to me the overall population from which the sample was obtained. I know the authors used the power calculation based on the three arms but what is the estimated population from which the sample was drawn.

Results

We included a total of 270 individuals (51% women) aged 18 to 49 years in marginalized regions, Falak od Din (role playing: 80), Masur (Control: 1100) and Poshteh (Lecture: 80) ---- I think that there is a typo on the Control group number. Is that supposed to be 110?

The mean score of knowledge was significantly lower in the control group before the intervention than the intervention groups (p=0.001) ---- This result is not too clear to me especially the phrase "before the intervention than the intervention groups".

Despite the lack of difference between the effectiveness of role-playing and lecture in improving attitude in the present study, studies have found that role-playing had a greater effect on improving knowledge and attitude than lecture, for instance, Manzari et al. (2015) found that more advanced training methods such as role-playing and multimedia could be more effective than the lecture in promoting knowledge and attitude ]19 ,18[. ---This sentence contain two referencing styles.
---

## [Decision Letter · Decision Letter 1]

6 Feb 2023

Improving the HIV-stigma in the Marginalized population in Khorramabad, Iran: a single-blinded randomized, controlled educational trial using role-playing and lecturing

PGPH-D-21-00852R1

Dear Dr Nasirian,

We are pleased to inform you that your manuscript 'Improving the HIV-stigma in the Marginalized population in Khorramabad, Iran: a single-blinded randomized, controlled educational trial using role-playing and lecturing' has been provisionally accepted for publication in PLOS Global Public Health.

Best regards,

Kavitha Saravu, MD, DNB, DTM&H (London)

Academic Editor

The reviewer comments have been addressed satisfactorily.

Reviewer Comments (if any, and for reference):

Reviewer's Responses to Questions

**Comments to the Author**

1. If the authors have adequately addressed your comments raised in a previous round of review and you feel that this manuscript is now acceptable for publication, you may indicate that here to bypass the “Comments to the Author” section, enter your conflict of interest statement in the “Confidential to Editor” section, and submit your "Accept" recommendation.

Reviewer #2: All comments have been addressed

2. Does this manuscript meet PLOS Global Public Health’s publication criteria? Is the manuscript technically sound, and do the data support the conclusions? The manuscript must describe methodologically and ethically rigorous research with conclusions that are appropriately drawn based on the data presented.

Reviewer #2: Yes

3. Has the statistical analysis been performed appropriately and rigorously?

Reviewer #2: I don't know

4. Have the authors made all data underlying the findings in their manuscript fully available (please refer to the Data Availability Statement at the start of the manuscript PDF file)?

Reviewer #2: Yes

5. Is the manuscript presented in an intelligible fashion and written in standard English?

Reviewer #2: No

6. Review Comments to the Author

Reviewer #2: (No Response)

7. PLOS authors have the option to publish the peer review history of their article (what does this mean?). If published, this will include your full peer review and any attached files.

**Do you want your identity to be public for this peer review?** For information about this choice, including consent withdrawal, please see our Privacy Policy.

Reviewer #2: **Yes: **Sneha Deepak Mallya
